# Novel Triterpenic Acid—Benzotriazole Esters Act as Pro-Apoptotic Antimelanoma Agents

**DOI:** 10.3390/ijms23179992

**Published:** 2022-09-01

**Authors:** Marius Mioc, Alexandra Mioc, Alexandra Prodea, Andreea Milan, Mihaela Balan-Porcarasu, Roxana Racoviceanu, Roxana Ghiulai, Gheorghe Iovanescu, Ioana Macasoi, George Draghici, Cristina Dehelean, Codruta Soica

**Affiliations:** 1Department of Pharmaceutical Chemistry, Faculty of Pharmacy, “Victor Babes” University of Medicine and Pharmacy, Eftimie Murgu Sq., No. 2, 300041 Timisoara, Romania; 2Research Centre for Pharmaco-Toxicological Evaluation, “Victor Babes” University of Medicine and Pharmacy, Eftimie Murgu Sq., No. 2, 300041 Timisoara, Romania; 3Department of Anatomy, Physiology, Pathophysiology, Faculty of Pharmacy, “Victor Babes” University of Medicine and Pharmacy, Eftimie Murgu Sq., No. 2, 300041 Timisoara, Romania; 4Institute of Macromolecular Chemistry ‘Petru Poni’, 700487 Iasi, Romania; 5Department of Otolaryngology, Faculty of Medicine, “Victor Babes” University of Medicine and Pharmacy, Eftimie Murgu Sq., No. 2, 300041 Timisoara, Romania; 6Department of Toxicology, Faculty of Pharmacy, “Victor Babes” University of Medicine and Pharmacy, Eftimie Murgu Sq., No. 2, 300041 Timisoara, Romania

**Keywords:** triterpenic acids, 1-hydroxybenzotriazole esters, melanoma, cytotoxicity, rtPCR, apoptosis, molecular docking

## Abstract

Pentacyclic triterpenes, such as betulinic, ursolic, and oleanolic acids are efficient and selective anticancer agents whose underlying mechanisms of action have been widely investigated. The introduction of N-bearing heterocycles (e.g., triazoles) into the structures of natural compounds (particularly pentacyclic triterpenes) has yielded semisynthetic derivatives with increased antiproliferative potential as opposed to unmodified starting compounds. In this work, we report the synthesis and biological assessment of benzotriazole esters of betulinic acid (BA), oleanolic acid (OA), and ursolic acid (UA) (compounds **1**–**3**). The esters were obtained in moderate yields (28–42%). All three compounds showed dose-dependent reductions in cell viability against A375 melanoma cells and no cytotoxic effects against healthy human keratinocytes. The morphology analysis of treated cells showed characteristic apoptotic changes consisting of nuclear shrinkage, condensation, fragmentation, and cellular membrane disruption. rtPCR analysis reinforced the proapoptotic evidence, showing a reduction in anti-apoptotic Bcl-2 expression and upregulation of the pro-apoptotic Bax. High-resolution respirometry studies showed that all three compounds were able to significantly inhibit mitochondrial function. Molecular docking showed that compounds **1**–**3** showed an increase in binding affinity against Bcl-2 as opposed to BA, OA, and UA and similar binding patterns compared to known Bcl-2 inhibitors.

## 1. Introduction

Drug discovery has been defined by the scientific community as a process meant to lead to the identification of potential new drugs; the path from the original idea to the marketable product is complex, long (12–15 years), and expensive (more than USD 1 billion) and involves multiple scientific domains, such as biology, chemistry, pharmacology, and toxicology [1]. Following the identification and validation of a target, drug discovery begins with the screening of various databases to identify hit molecules worthy of clinical development. Both synthetic and natural-based drugs have limitations (i.e., chemical instability, poor pharmacokinetic properties, and unwanted side effects), which can be overcome by adopting a hybrid approach, the semisynthetic pathway that consists in the derivatization of a natural compound through chemical modulation to achieve optimized pharmacokinetic and pharmacodynamic properties. In addition, the preparation of semisynthetic derivatives may pose fewer technical challenges compared to fully synthetic drugs.

Pentacyclic triterpenoids are active phytochemicals found in higher plants, which contain a main skeleton generally represented by lupane, oleanane, and ursane, and are formed through biosynthesis from isoprene units [2]; they exert a multitude of pharmacological effects, including anticancer and anti-inflammatory, through various molecular mechanisms. Although many biological activities have been identified for pentacyclic triterpenes, the most investigated effects by far were their anticancer and chemopreventive effects [3].

Among pentacyclic triterpenes, betulinic (BA), ursolic (UA), and oleanolic (OA) acids have been identified as efficient and selective anticancer agents whose underlying mechanisms of action have been widely investigated. Despite their large plethora of pharmacological effects, triterpenic acids exhibit very low water solubility, which has been correlated to their poor oral bioavailability. In addition to various technological approaches to solve this issue, such as cyclodextrin complexation or nano-formulations, different chemical modulations have been conducted on the compounds’ molecules, leading to the synthesis and assessment of numerous semisynthetic derivatives. The main molecular positions targeted by chemical modulations were the C3 hydroxy, C28 carboxylic acid, and ring A whose transformations led to numerous derivatives that exhibited stronger antitumor activities than both the parent compound and marketed drugs used as the positive control [4,5,6]. C28 triazolyl derivatives of BA were synthesized by Dangroo et al. by means of Huisgen 1,3-dipolar cycloaddition with some compounds acting highly effective against several cancer cell lines; the study also showed that the introduction of non-polar alkyl groups at the C-28 position caused a significant loss of the antitumor activity [7]. A similar approach led to a series of novel oleanolic acid-coupled 1,2,3-triazole derivatives, some of which revealed excellent anticancer activities [8]. The cycloaddition reaction was used on several alkynes and azides previously obtained from betulinic acid and allylic alcohols to synthesize new triazole derivatives with anticancer properties; the resulting compounds revealed promising antiproliferative effects against pancreatic and breast cancers. An important finding of the study was that the triazole moiety may serve as a key pharmacophore [9]. In fact, 1,2,3-triazole is an important pharmacophore in medicinal chemistry that can form hydrogen bonds with biological targets, thus intensifying the overall pharmacological effect [10].

1-Hydroxybenzotriazole is a widely-used coupling reagent along with carbodiimides used mostly in amide and ester synthesis [11,12]. While this reagent is mainly used to form intermediate active esters of various natural compounds that further react to form final amides or esters, several studies reported the biological assessment of various 1-hydroxybenzotriazole-triterpene esters, some of which exhibited higher biological effects as opposed to unmodified triterpenes [13,14,15,16,17]. Therefore, 1-hydroxybenzotriazole is a reagent that can be easily used to obtain triazole-containing triterpene esters, with potential increased biological potential.

In the current study, we propose the synthesis and antiproliferative biological assessment of BA, OA, and UA-1-hydroxybenzotriazole esters against melanoma. Moreover, this work attempts to elucidate the mechanism of action of the novel synthesized compounds correlated with their antiproliferative effects.

## 2. Results

### 2.1. Chemistry

Compounds **1**–**3** were obtained using the *N*,*N*′-dicyclohexylcarbodiimide (DCC)-coupled esterification reactions of BA, OA, and UA using 1-hydroxybenzotriazole (HOBt) (Figure 1). DCC reacts with the COOH group of the triterpenic acid, forming an iso-acylurea intermediate, which is converted to esters **1**–**3** upon adding the HOBt. The reaction was periodically checked for completion using TLC (hexane/ethyl acetate 1:1). After 4 h, small traces of unreacted triterpene were noticeable along with unreacted excess HOBt. The products were purified using column chromatography (hexane/ethyl acetate 3:1), which easily removed traces of unreacted products and the dicyclohexylurea by-product formed in the process.

### 2.2. In Vitro Toxicological Evaluation of Compounds ***1***, ***2***, and ***3*** in HaCaT and A375 Cell Lines

The effects of **1**, **2,** and **3** and their corresponding acids (BA, OA, and UA) on the viability of healthy human keratinocytes HaCaT and human melanoma A375 cells were evaluated following a treatment period of 24 h by the means of the Alamar Blue assay. In HaCaT cell lines, compounds **1** and **2** did not exert any cytotoxic effects on normal cells when applied in concentrations up to 25 μM (Figure 2A,B). However, at the highest concentration (50 μM), both compounds, and their corresponding acids (BA and OA) significantly decreased cell viability as follows: 88.2% viable cells after treatment with compound **1**, 87.1% BA, 87% viable cells after treatment with compound **2**, and 86.6% OA (Figure 2A,B). No obvious evidence of cytotoxicity was observed on HaCaT cells after 24 h of treatment with UA and with different concentrations of compound **3,** representing excellent cytocompatibility of this compound with normal cells, even at high concentrations (Figure 2C). In A375 melanoma cells, cell viability was significantly reduced when compounds **1** and **2** were tested at 25 μΜ (81.25% and 87.4% vs. the control—100%) and 50 μΜ (69.8% and 62.5% vs. the control—100%). At 50 μM, the original acids tested alone, BA and OA, also significantly decreased cell viability (75% and 74.8% vs. the control—100%); however, to a lesser extent than the newly synthesized compounds **1** and **2** (Figure 2A,B). Furthermore, UA alone and compound **3** only at 50 μΜ decreased cell viability to 85% and 77% vs. the control—100% (Figure 2C).

Cell morphological assessment was conducted to identify the underlying antiproliferative mechanism; changes in cell morphology consisting of floating round cells, reduced confluence, and loss of cellular adherence are clear indicators of cytotoxicity. These aspects also appear in the HaCaT treated cell line but to a far lesser degree as opposed to the treated melanoma cells (Figure 3 and Figure 4). Cells were further submitted to nuclear staining using the 4,6′-diamidino-2-phenylindole (DAPI) dye, which binds to the DNA and causes blue fluorescence under UV light. The results are depicted in Figure 5 and Figure 6. A more in-depth discussion regarding these results is highlighted in Section 3: “3. Discussion”.

### 2.3. Quantitative Real-Time PCR

To further establish the pro-apoptotic effects of compounds **1**–**3** on A375 melanoma cells, a quantitative real-time PCR was employed to determine gene expression variations of anti-apoptotic Bcl-2 and pro-apoptotic Bax. Measurements were recorded on A375 cells treated with test compounds after 24 h of incubation. The effect of each compound was recorded at the minimum concentration at which each compound showed a significant reduction in cell viability (**1, 2**—25 μM, **3**—50 μM). Results show that all compounds induce an upregulation in the expression of the pro-apoptotic Bax gene and downregulation of anti-apoptotic Bcl-2 (Figure 7).

### 2.4. High-Resolution Respirometry Studies

Over time, an extensive amount of literature has been compiled on mitochondrial involvement in promoting cancer development and progression and, thus, compounds that are able to selectively target cancer cell mitochondrial functions are viewed as promising tools in cancer therapeutics [18,19]. Therefore, to evaluate the effect on the mitochondrial functions of HaCaT healthy cells and A375 melanoma cells, the novel compounds **1**, **2,** and **3** were tested by high-resolution respirometry.

The obtained results show that compounds **1**, **2,** and **3** do not significantly influence mitochondrial respiration of HaCaT cells when used in concentrations found to significantly decrease cell viability of A375 melanoma cells (25 μM for compounds **1** and **2** and 50 μM for compound **3**) (Figure 8, Figure 9 and Figure 10).

In A375 melanoma cells, all three compounds were able to significantly inhibit mitochondrial function. In detail, compounds **1**, **2,** and **3** decreased routine respiration, suggesting a decrease in cellular mitochondrial ATP demand and energy turnover (Figure 8, Figure 9 and Figure 10). The decrease of State 2_CI_ and State 4_CI+CII_ reveals that only compounds **1** and **2** can lower the residual respiration that compensates for the proton leak in the non-phosphorylating resting state when ATP synthase is not active (Figure 8 and Figure 9), whereas compound **3** did not significantly influence this mitochondrial rate (Figure 10). Upon further exploring the meaning of these modifications, the link between proton leak (State 2_CI_ and State 4_CI+CII_) and reactive oxygen species (ROS) production must be addressed. Despite contradictory data, it seems that uncoupling (i.e., increase of State 2_CI_ and State 4_CI+CII_ rates due to the increase of the proton leak across the inner mitochondrial membrane) decreases mitochondrial ROS production. In the current setting, the observed decrease of State 2_CI_ and State 4_CI+CII_ may suggest that compounds **1** and **2** can increase, whereas compound **3** does not influence mitochondrial ROS production. Analyzing further, all compounds significantly decreased active respiration (OXPHOS_CI_ and OXPHOS_CI+CII_) and the maximal capacity of the electron transport system (ETS_CI_ and ETS_CI+CII_), suggesting that all compounds inhibit mitochondrial respiration (Figure 8, Figure 9 and Figure 10).

Taken together, the presented data suggest that compounds **1** and **2** at 25 μM and compound **3** at 50 can μM-selectively induce apoptosis via the intrinsic pathway and inhibit mitochondrial function in A375 melanoma cells, without exhibiting any toxic effect on HaCaT healthy cells.

### 2.5. Molecular Docking

Following our biological assessment results for compounds **1**–**3**, we employed molecular docking to analyze the binding affinity and interaction mode of these structures against Bcl-2. We also docked the parent structures, BA, OA, and UA, respectively, to comparatively assess if there was an increase in the theoretical affinity towards Bcl-2 due to the employed structural changes. The obtained docking results are presented in Table 1 and the binding interactions of compounds **1**–**3** with Bcl-2 are depicted in Figure 11. A detailed interpretation of these results is discussed in Section 3 (Discussion).

## 3. Discussion

Triterpenes represent a large group of natural products with various biological effects, of which, several hundred new compounds are reported each year worldwide; although neither natural nor semisynthetic triterpenes have entered the anticancer arsenal, the oleanane-derived bardoxolone methyl-bearing chemical modulations in both C3 and C28 positions is currently in a clinical trial for its potential use against both solid tumors and lymphomas [20]. The triazole heterocycle, either 1,2,3-triazole or 1,2,4-triazole, has been extensively used in drug discovery due to its low occurrence in nature as well as its straightforward synthesis. The triazole ring provides strong hydrogen bonding within the in vivo environment accompanied by high chemical stability and rigidity, which enable its use as a pharmacophore in numerous therapeutical applications [21]. The introduction of a 1,2,3-triazole moiety through the esterification of the carboxyl group with HOBt is a feasible approach that was conducted on other pentacyclic triterpenes, such as esculentoside A, where some of the obtained compounds were more potent and less toxic than the parent compound [16]. Using the same chemical pathway, Schwarz et al. prepared the 1H-benzotriazolyl ester of the pentacyclic glycyrrhetinic acid, which exhibited a strong selective inhibitory activity against butyrylcholinesterase [17], thus acting as an anti-Alzheimer’s agent. The chemical synthesis applied in the current paper led to the preparation of three benzotriazolyl esters of the three triterpenic acids (BA, UA, and OA, respectively); the synthesis was confirmed by the physicochemical assessment of the resulting compounds whose structure was confirmed by means of ^1^H- and ^13^C-NMR, FTIR, and LC-MS. All recorded spectra are available in the Appendix A of this article.

The benzotriazolyl esters were tested in terms of cytotoxic activities in A375 human melanoma cells and HaCaT healthy keratinocytes by employing the Alamar Blue cell viability test; cell viability translates into the cells’ ability to perform a specific function or to engage in the mitotic process. The assessment of cell viability is necessary either to establish cell morphological integrity or to monitor cell reactions under various stimuli including toxins or other pharmacologically active compounds [22]. In the current study, all three semisynthetic compounds exerted a significant dose-dependent cytotoxic effect in melanoma cells when used in concentrations above 25 μΜ; their biological activities exceeded in all cases the ones recorded for the parent compounds. However, the cytotoxic effects of UA and its derivative manifested only when the highest concentration (50 μΜ) was applied while for the other two triterpenic acids and their esters the biological activity occurred at 25 μΜ; in addition, the decrease in cell viability was more pronounced after the application of BA and OA and their respective derivatives by comparison to UA and its ester in similar concentrations (Figure 2). Simultaneously, all compounds were also tested on healthy HaCaT cells to assess their selectivity level; the application of 25 μΜ samples did not induce any changes in cell viability compared to control regardless of the tested compound. However, when the concentration increased at 50 μΜ, cytotoxic effects occurred for BA and OA as well as their respective derivatives; for UA and its ester, no cytotoxic activity was recorded (Figure 2). In agreement with our data, BA was previously reported to be more effective as an anti-melanoma agent at lower concentrations than either OA or UA; Isaković-Vidović et al. emphasized that its inhibitory activity against the WM-266-4 metastatic melanoma cell line was dose-dependent regardless of the time of incubation [23]. Our own previous studies revealed that BA [24], OA, and UA [25] exert dose-dependent cytotoxic effects in A375 human melanoma cells. Similar to the current study, the three triterpenic acids showed no signs of cytotoxicity in normal mesenchymal cells when used in low concentrations (up to 25 μΜ); however, at a higher concentration, BA induced a dose-dependent decrease in cell viability [24]. When compared to parent compounds, the newly synthesized esters revealed stronger cytotoxic activities combined with higher selectivity; one such ester was obtained by Khan et al. who synthesized a library of BA derivatives and tested their cytotoxic effects against four different cancer cell lines [26]. All compounds displayed different growth inhibition effects with maximum inhibitory activity exerted by 28{1N(4-fluoro phenyl)-1H-1,2,3-triazol-4-yl} methyloxy betulinic ester (compound C12); the authors attributed its strong cytotoxic effect to the presence of the F-bearing electron-withdrawing functional group attached to the triazole moiety which endows the entire molecule with increased lipophilicity and thus enables cell penetration. A library of 1,2,3-triazole-substituted esters of oleanolic acid was designed and synthesized by Wei et al. who determined that several compounds displayed excellent anticancer activity against several tumor cell lines, in particular HT1080 fibrosarcoma cells [8]. The modulation of the C28 position in the molecule of UA was conducted by Wang et al. resulting in the synthesis of several ester derivatives bearing various moieties, including 1,2,3-triazole; the 1,2,3-triazole derivatives induced a moderate decrease in cell viability when tested against Hela and MKN45 cell lines [27]. Collectively, data show that the modulation of the C28 position in the molecule of triterpenic acids in the form of benzotriazolyl esters may lead to the synthesis of more effective and selective antimelanoma agents; to the best of our knowledge, this is the first report on the synthesis of such compounds followed by their assessment as treatment against melanoma.

Numerous reports, including those above, have revealed that the main molecular mechanism involved in the anticancer activity of triterpenic acids and their derivatives consists of apoptosis induction through the regulation of pro- and anti-apoptotic genes. Apoptosis is a biological process of extensive importance, which consists of the orchestrated death of damaged cells followed by phagocytosis; aberrant apoptosis may lead to abnormal cell proliferation and severe diseases, such as cancer, and also stands as a reason for the development of anticancer drugs resistance [28]. The morphological assessment of cells was conducted to identify the underlying antiproliferative mechanism; changes in cell morphology in the form of floating round cells, reduced confluence, and loss of cellular adherence are clear indicators of cytotoxicity (Figure 4). Cells were further submitted to nuclear staining using DAPI dye, which binds to the DNA and causes blue fluorescence under UV light. According to the reported data, HaCaT cells showed signs of apoptosis, i.e., nuclear shrinkage or fragmentation, only when the highest concentrations of BA and OA were used (Figure 5); no signs of apoptosis or necrosis were recorded for UA or its benzotriazole ester regardless of the tested concentration. BA and OA esters, compounds **1** and **2**, were tested at 25 μΜ, a concentration which decreased the cell viability in the A375 cell line but not in the HaCaT cell line; the results show that, at this concentration, **1** and **2** did not induce any signs of apoptosis in HaCaT (Figure 5) cell line. In A375 melanoma cells, apoptotic changes occurred after the treatment with BA, OA, UA (50 μΜ), and their esters **1**, **2** (25 μΜ), and **3** (50 μΜ) as indicated by nuclear shrinkage, which characterizes the onset of the process, chromatin condensation, and nuclear fragmentation. All changes were confirmed by using staurosporine as a positive control (Figure 6). Our group reported similar results for BA [24] as well as for the two isomers, UA and OA [25]; there are also other studies that confirm the induction of apoptosis as an anticancer mechanism for pentacyclic triterpenes [29]. In addition, several esters of the C28 carboxylic group in the molecule of pentacyclic triterpenoids revealed apoptotic effects [30,31,32,33]. Another mechanism by which triterpenes can exert their cytotoxic effects is to impair cell membrane integrity due to their membranotropic potential. This effect is commonly present amongst triterpene glycosides being closely related to the compounds’ chemical structure and related hydrophobicity [34]. The literature suggests that in order for these types of triterpene glycosides to modify the cell membrane, the presence of a linear tetrasaccharide chain is necessary [34]. Moreover, the membranotropic-induced toxicity of triterpenes visibly affects cancer and healthy cells alike, as reported by a recent study [35]. The authors revealed that while the tested compounds induced significant cytotoxicity in various cancer cells, they also suppressed the viability of normal cells when tested at the same concentrations [35]. Regarding compounds **1**–**3,** we concluded that this is not the primary cause because this effect is closely correlated with the hydrophobicity of the compounds, which is very similar. That could have translated into a cytotoxic effect of the compounds on healthy cells as well, an effect that was not present. Moreover, compounds **2** and **3** are highly similar in terms of structure and hydrophobicity but have different effects on cells, thus showing that these effects are not solely influenced by the structures’ hydrophobic nature alone. Another recent work showed that betulonic acid, a betulinic acid derivative, affects mitochondrial function but has no effect on membrane permeability [36]. Collectively, results suggest that both natural triterpenic acids as well as their respective ester derivatives exert dose-dependent cytotoxic effects in melanoma cells through apoptosis induction. In addition, low concentrations of both natural and semisynthetic compounds induced high selectivity in tumor vs. healthy cells while higher concentrations triggered a moderate level of cytotoxicity; similar results were reported for OA that caused the occurrence of apoptotic features (i.e., altered cell morphology and DNA) in HaCaT cells in a dose-dependent manner but with low overall cytotoxicity [37]. Interestingly, the current study revealed no decrease in HaCaT cell viability for either UA or its ester regardless of concentration while a previous study reported such an antiproliferative effect in a concentration- and time-dependent manner [38].

The apoptotic process can develop by intrinsic or extrinsic pathways; the intrinsic or mitochondrial pathway is triggered by internal cellular stress and involves the balanced actions of pro-apoptotic (Bax, Bak) and anti-apoptotic (BCL-2, BCL-X, BCL-w, MCL-1, BFL-1/A1) proteins. Betulinic acid was shown to increase the expression of Bax while downregulating the anti-apoptotic protein Bcl-2, thus inducing apoptosis in PANC-1 and SW1990 pancreatic cells [39]. Additionally, oleanolic acid [40] as well as its isomer, ursolic acid [41], upregulated Bax and Caspase-9 genes and downregulated Bcl-2 in various cancer cells. It was only natural to hypothesize that their respective triazole ester derivatives will act in a similar manner given the fact that the previously mentioned 28{1N (4-fluoro phenyl)-1H-1,2,3-triazol-4-yl} methyloxy betulinic ester clearly decreased Bcl-2/Bax ratio and activated caspase-9, thus inducing intrinsic apoptosis in HL-60 leukemia cells [26]. As predicted, compounds **1**, **2,** and **3** downregulated mitochondrial Bcl-2 and upregulated Bax gene expression, thus suggesting a potent induction of mitochondria-associated apoptosis in A375 melanoma cells (Figure 7).

The investigation of the apoptotic mechanism induced by pentacyclic triterpenic acids revealed that they can act as mitocans (mitochondria targeting anticancer agents), which trigger apoptosis through the intrinsic pathway [42]. To investigate the finer mechanisms of action of the three newly synthesized esters, we performed high-resolution respirometry studies in A375 human melanoma cells as well as in HaCaT normal cells; previous studies using the same protocol revealed that BA stimulation triggered an overall inhibition of cellular respiration in A375 melanoma cells as indicated by the inhibition of basal and active respiration, as well as of maximal uncoupled respiration [24]. Even under normoxic conditions, cancer cells display much higher rates of glycolysis to produce ATP compared to normal cells, which rely on oxidative phosphorylation (OXPHOS) [43]. However, despite their general high glycolytic rates, tumors display small subpopulations of slow-cycling cells, which are dependent on mitochondrial respiration and suggest a common metabolic program across various tumors [44].

Recently, it has been shown that certain subsets of melanomas displaying primary resistance toward targeted therapies seem to rely more on mitochondrial respiration compared to glycolysis [45]. Metabolic plasticity between glycolysis and OXPHOS has been reported for melanoma cells, which provides cell adaptability to the environment and, implicitly, pathways of chemoresistance; in addition, melanoma progression seems to depend on the simultaneous upregulation of both OXPHOS and glycolysis [46]. Therefore, considering that mitochondria provide up to 75% of ATP demands in cancer cells, tumor cells with defective OXPHOS develop high sensitivity to cytotoxic drugs [24]; currently, the inhibition of mitochondrial respiration is regarded as a promising strategy to overcome drug resistance.

In normal HaCaT cells, compounds **1**, **2,** and **3** do not significantly influence mitochondrial respiration when used in concentrations proven effective against A375 melanoma cells as indicated by reduced cell viability. In contrast, the semisynthetic compounds were revealed to inhibit mitochondrial function and decrease routine respiration, which translates into a decrease in cellular mitochondrial ATP demand and energy turnover (Figure 8, Figure 9 and Figure 10). Recently, cancer progression and resistance have been associated with increased mitochondrial uncoupling [47,48]. As previously reported by our group, BA does not exert the effects of a classical uncoupler by inducing mitochondrial dysfunction without increasing the proton transport across the inner mitochondrial membrane [24]; i.e., a compound able to increase the proton transfer across the inner mitochondrial membrane and, thus, dissociate between the generation of mitochondrial membrane potential and its utilization to produce mitochondrial ATP, an effect observed by increased State 2_CI_ and State 4_CI+CII_ respiratory rates. BA also showed a dose-dependent decrease in mitochondrial membrane potential [24]. Accordingly, compounds **1** and **2** were able to reduce State 2_CI_ and State 4_CI+CII_ in A375 cell lines at 25 μM, indicating that they induce a decrease in the oxygen consumption in the non-phosphorylating resting state (when ATP synthase is not active) and also a decrease of the proton transport across the inner mitochondrial membrane; by contrast, compound **3** (UA derivative) did not significantly alter this parameter at 50 μM, thus suggesting that **3** can trigger mitochondrial dysfunction via another mechanism that does not imply the proton transfer across the inner mitochondrial membrane (Figure 8, Figure 9 and Figure 10). It has long been demonstrated that mitochondrial oxidative phosphorylation leads to ROS generation, which can be regulated by proton transfer (leak) across the mitochondrial inner membrane; the relationship between the increasing proton leak (uncoupling) and ROS generation goes both ways in a feedback loop: the proton leak decreases the production of ROS and, in turn, ROS can trigger the proton leak [49]. Therefore, considering the current data, the recorded decrease of State 2_CI_ and State 4_CI+CII_ may suggest that compounds **1** and **2** can increase mitochondrial ROS production; compound **3** did not produce evidence of such an effect. Natural triterpenoids have been characterized as ROS modulators, thus offering the possibility to generate derivatives with high selectivity on ROS regulation [50]. The stimulatory activity of compounds **1** and **2** on ROS generation may partly explain their apoptotic activity in melanoma cells. Furthermore, all compounds significantly decreased active respiration (OXPHOS_CI_ and OXPHOS_CI+CII_) and the maximal capacity of the electron transport system (ETS_CI_ and ETS_CI+CII_), thus inhibiting active mitochondrial respiration and diminishing the maximal respiratory capacity, the latter being considered a strong indicator of mitochondrial dysfunction [51].

Molecular docking is one of the most successful structure-based in silico methods widely used in drug discovery, which enables the identification of new active drugs through accurately predicting their interactions with biological targets; the process begins with the prediction of the molecular orientation of a drug molecule within a potential receptor followed by the estimation of their complementarity using a score function [52]. At first glance, docking results show that compounds **1**–**3** recorded a noticeable decrease in binding energy and a subsequent increase in affinity towards Bcl-2 as compared to the parent compounds BA, OA, and UA, respectively. We can also observe that the same compounds **1**–**3** exhibit relatively close docking scores compared to the native ligand (NL). This aspect is closely related to the undergone structural change and the correlated interactions between these compounds and the active site of Bcl-2. Bcl-2 has a large hydrophobic groove where its BH domain containing ligands bind [53]. The NL of Bcl-2, used as docking control in this study, is a phenyl-pyrazole derivative containing a tetrahydroisoquinoline (THIQ) moiety. This compound interacts with hydrophobic residues within the same cleft where Bak, Bad, or Bim peptides bind [54] but also has a unique feature. The THIQ radical interacts with another pocket where the peptide ligands do not interact (Asp70, Phe71). In our case, compound **1** shows the highest similarity with the NL regarding interactions with amino acid residues (Asp70, Phe71, Met74), with the benzotriazole ring acting in the same way as the THIQ ring from the NL structure (Figure 11). While compound **3** also forms hydrophobic interactions with Ala108 and Arg105 and has a flipped orientation compared to compound **1** (Figure 11), compound **2** behaves in a slightly different manner. In this case, the benzotriazole ring strongly interacts with Arg105 through multiple hydrogen bonds (HB) while at the other pole the structure interacts with Asp70 as well, making the molecule heavily anchored within the binding pocket (Figure 11). Nevertheless, the benzotriazole ring could represent a valuable option in the design of highly potent triterpene-based Bcl-2 inhibitors.

## 4. Materials and Methods

### 4.1. Chemistry

#### 4.1.1. General

Betulinic acid (BA), oleanolic acid (OA), ursolic acid (UA), 1-hydroxybenzotriazole hydrate (HOBt), *N*,*N*′-dicyclohexylcarbodiimide (DCC), *N*,*N*-dimethylformamide (DMF) and all other necessary solvents were commercially available products (Merck KGaA, Darmstadt, Germany) and were further used without any additional purification. Melting points were obtained using a Biobase melting point apparatus (Biobase Group, Shandong, China); thin-layer chromatography (TLC) was achieved using silica gel-coated plates 60 F254 (Merck KGaA, Darmstadt, Germany) and various ratios of hexane: ethyl acetate, as eluents. FTIR spectra of the tested compounds were recorded on a Shimadzu IR Affinity-1S spectrophotometer in the range of 400–4000 cm^−1^ (4 cm^−1^ resolution) using potassium bromide pelletization. The 1D and 2D NMR spectra were recorded on a Bruker Avance NEO 400 MHz Spectrometer equipped with a 5 mm probe for the direct detection of H, C, F, Si. All the spectra were recorded at room temperature using DMSO-d6 as solvent and standard parameter sets provided by Bruker. The chemical shifts are reported as δ values (ppm) relative to the solvent residual peak (2.51 ppm for ^1^H and 39.5 ppm for ^13^C). The assignment of the peaks in the ^1^H and ^13^C NMR spectra was done using information from additional 1D and 2D NMR experiments: ^13^C-DEPT135, H,H-COSY (Correlation Spectroscopy), H,C-HSQC (Heteronuclear Single Quantum Coherence), and H,C-HMBC (heteronuclear multiple bond correlation). LC/MS analysis was conducted on an Agilent 6120 Quadrupole LC/MS system (Santa Clara, CA USA) equipped with a UV detector, ESI ionization source, and a Zorbax Rapid Resolution SB-C18 (1.8 μm; 50 × 2.1 mm) column. LC/MS spectra of all samples were carried out using methanolic solutions of the compounds tested. All samples were analyzed using LC/MS grade methanol (Merck, Darmstadt, Germany) as an isocratic mobile phase, at a flow rate of 0.4 mL/min, temperature of 25 °C, and l = 250 nm. Mass spectra were recorded in the positive ion mode using optimized ESI parameters: nitrogen nebulizer pressure, 35 psi, nitrogen drying gas temperature, 250 °C, flow rate at 12 L/min, and capillary voltage set at 3000 V, in the Scan mode.

#### 4.1.2. General Synthesis Procedure for Compounds **1**–**3**

A round bottom flask, previously equipped with a magnetic stirrer, was loaded with 0.5 mmol of triterpenic acid, dissolved in 5 mL DMF. DCC (1.5 eq) was subsequently added to the solution, the mixture was stirred for 30 min, after which 1.5 eq of HOBt was added. The reaction was kept under continuous stirring, at room temperature for 4 h. The final solution was precipitated in water, filtered, and the obtained solid was purified using column chromatography (hexane:ethyl acetate 3:1). All recorded spectra are available in the Appendix A of this article .

#### 4.1.3. 1H-Benzotriazole-1-yl (3β) 3-hydroxy-20(29)-lupaene-28-oate (1)

White powder, yield 28%, Mp 136–142 °C; Rf 0.72 (hexane/ethyl acetate 1:1); **^1^H NMR** (400.13 MHz, DMSO-d6, δ, ppm): 8.18 (d, J = 8.4 Hz, 1H, H5′), 7.72 (t, J = 7.4 Hz, 1H, H7′), 7.64 (d, J = 8.3 HZ, 1H, H8′), 7.55 (t, J = 7.6 Hz, 1H, H6′), 4.73 (s, 1H, H29a), 4.61 (s, 1H, H29b), 4.28 (d, J = 5.2 Hz, 1H, OH), 2.98 (m, 1H, H3), 2.83 (m, 1H, H19), 2.51 (m, 1H, H16a, overlapped with solvent residual peak), 2.34 (m, 1H, H22a), 2.08 (td, J = 2.7 Hz, J = 11.9 Hz, 1H, H13), 1.91–1.96 (m, 2H, H21a, H22b), 1.81–1.87 (m, 2H, H16b, H9), 1.70 (s, 3H, H30), 1.62–1.65 (m. 2H, H12a, H21b), 1.24–1.57 (m, 12H, H1a, H2, H6, H7, H11a, H15, H18), 0.98–1.1 (m, 5H, H11b, H12b, H26), 0.82–0.9 (m, 7H, H1b, H23, H27), 0.76 (s, 3H, H25), 0.67–0.69 (m, 4H, H5, H24); **^13^C NMR** (100.6 MHz, DMSO-d6, δ, ppm:): 171.9 (C28), 149.1 (C20), 142.8 (C4′), 129.4 (C7′), 128.3 (C9′), 125.3 (C6′), 120.0 (C5′), 110.3 (C29), 108.3 (C8′), 76.7 (C3), 56.5 (C17), 54.8 (C5), 49.8 (C18), 49.0 (C9), 46.4 (C19), 42.0 (C14), 40.2 (C8), 38.4 (C4), 38.2 (C1), 38.1 (C13), 36.6 (C10), 35.7 (C22), 33.8 (C7), 30.4 (C16), 29.7 (C21), 29.5 (C15), 28.0 (C23), 27.1 (C2), 28.4 (C12), 20.3 (C11), 18.9 (C30), 17.9 (C6), 15.8 (C24), 15.7 (C27), 15.6 (C25), 14.4 (C26). **FTIR** [KBr] (cm^−1^): 3446 cm^−1^ (O-H stretch); 2945, 2872 (C-H stretch); 1809, 1240, 1064 (C=O, C-C-O, O-C-C stretch); 1737 (N-N stretch); 1242 (=C-N stretch); 742,765 (sp^2^ ar. C-H bend). **LC-MS** (ESI): *m*/*z* = 574.4 [M+ H+]+, calculated M = 573.81.

#### 4.1.4. 1H-Benzotriazole-1-yl (3β) 3-hydroxyolean-12-en-28-oate (2)

White powder, yield 38%, Mp 138–146 °C; Rf 0.74 (hexane/ethyl acetate 1:1); **^1^H NMR** (400.13 MHz, DMSO-d6, δ, ppm): 8.15 (d, J = 8.4 Hz, 1H, H5′), 7.69 (t, J = 7.6 Hz, 1H, H7′), 7.61 (d, J = 8.1 Hz, 1H, H8′), 7.53 (t, J = 7.6 Hz, 1H, H6′), 5.30 (s, 1H, H12), 4.31 (d, J = 5.1 Hz, 1H, OH), 2.98–3.04 (m, 1H, H3), 2.85–2.89 (m, 1H, H18), 2.28–2.35 (m, 1H, H11a) 2.09 (td, J = 3.8 Hz, J = 13.8 Hz, 1H, H15a), 1.97–2.00 (m, 1H, H15b), 1.71–1.92 (m, 6H, H11b, H16, H19a, H22), 1.46–1.60 (m, 8H, H1a, H2, H6a, H7a, H9, H21), 1.28–1.39 (m, 4H, H4, H6b, H7b), 1.16–1.23 (m, 4H, H19b, H27), 0.96 (s, 6H, H29, H30), 0.88–0.91 (m, 4H, H1b, H23), 0.85 (s, 3H, H25), 0.75 (s, 3H, H26), 0.69–0.72 (m, 4H, H5, H24); **^13^C NMR** (100.6 MHz, DMSO-d6, δ, ppm:): 173.5 (C28), 142.7 (C4′), 142.0 (C13), 129.3 (C7′), 128.1 (C9′), 125.2 (C6′), 123.1 (C12), 120.0 (C5′), 108.5 (C8′), 76.7 (C3), 54.7 (C5), 47.1 (C17), 46.9 (C9), 44.8 (C19), 41.4 (C14), 41.2 (C18), 38.9 (C8), 38.3 (C20), 38.0 (C1), 36.5 (C10), 32.8 (C21), 32.4 (C7), 32.3 (C29), 31.8 (C22), 30.2 (C15), 28.2 (C4), 27.5 (C23), 26.9 (C2), 25.4 (C27), 23.2 (C30), 22.9 (C16), 22.5 (C11), 17.9 (C6), 16.8 (C26), 16.0 (C24), 15.1 (C25). **FTIR** [KBr] (cm^−1^): 3441 (O-H stretch); 2939, 2862 (C-H stretch); 1811, 1246, 1049 (C=O, C-C-O, O-C-C stretch); 1734 (N-N stretch); 1246 (=C-N stretch); 746,783 (sp^2^ ar. C-H bend). **LC-MS** (ESI): *m*/*z* = 574.4 [M + H+]+, calculated M = 573.81.

#### 4.1.5. 1H-Benzotriazole-1-yl (3β) 3-hydroxyurs-12-en-28-oate (3)

White powder, yield 42%, Mp 138–144 °C; Rf 0.76 (hexane/ethyl acetate 1:1); **^1^H NMR** (400.13 MHz, DMSO-d6, δ, ppm): 8.14 (d, J = 8.4 Hz, 1H, H5′), 7.68 (t, J = 7.6 Hz, 1H, H7′), 7.50–7.57 (m, 2H, H6′, H8′), 5.30 (s, 1H, H12), 4.31 (d, J = 5.0 Hz, 1H, OH), 2.99–3.04 (m, 1H, H3), 2.29–2.35 (m, 2H, H16a, H18), 2.13 (d, J = 12.6 Hz, 1H, H22a), 1.98–2.05 (m, 2H, H15a, H22b), 1.81–1.95 (m, 3H, H11, H16b), 1.62 (d, J = 10.7 Hz, 1H, H21a), 1.45–1.55 (m, 9H, H1, H2, H6a, H7a, H9, H19a, H20a), 1.26–1.36 (m, 4H, H6b, H7b, H15b, H21b), 1.05–1.13 (m, 4H, H19b, H27), 0.97 (d, J = 6.2 Hz, 3H, H30), 0.91–0.93 (m, 4H, H20b, H24), 0.86–0.87 (m, 6H, H25, H29), 0.80 (s, 3H, H26), 0.69–0.72 (m, 4H, H5, H23); **^13^C NMR** (100.6 MHz, DMSO-d6, δ, ppm:): 173.1 (C28), 142.7 (C4′), 136.8 (C13), 129.2 (C7′), 128.0 (C9′), 126.2 (C12), 125.2 (C6′), 120.0 (C5′), 108.4 (C8′), 76.7 (C3), 54.7 (C5), 52.4 (C18), 48.6 (C17), 46.9 (C9), 41.8 (C14), 39.2 (C8), 38.8 (C4), 38.3 (C20), 38.2 (C1), 38.0 (C19), 36.4 (C10), 36.0 (C22), 32.7 (C7), 29.8 (C21), 28.2 (C24), 27.7 (C15), 26.9 (C2), 23.7 (C16), 23.0 (C27), 22.9 (C11), 20.7 (C30), 17.9 (C6), 17.1 (C26), 16.7 (C29), 16.0 (C23), 15.2 (C25); **FTIR** [KBr] (cm^−1^): 3446 (O-H stretch); 2927, 2872 (C-H stretch); 1805, 1242, 1047 (C=O, C-C-O, O-C-C stretch); 1737 (N-N stretch); 1242 (=C-N stretch); 742,781 (sp^2^ ar. C-H bend). **LC-MS** (ESI): *m*/*z* = 574.4 [M + H+]+, calculated M = 573.81.

### 4.2. Cell Culture

The immortalized human keratinocytes (HaCaT) and human melanoma cell lines (A375) (ATCC^®^ CRL-1619™) were purchased from CLS Cell Lines Service GmbH (Eppelheim, Germany) and American Type Culture Collection (ATTC, Łomianki, Poland), respectively. The cells were received as frozen items and were stored in liquid nitrogen. Both cell lines were seeded and propagated in Dulbecco’s Modified Eagle Medium (DMEM) high glucose, supplied with a mixture of 10% fetal bovine serum (FCS) and 1%, a combination of 100 U/mL of penicillin and 100 U/mL of streptomycin (Sigma-Aldrich, Munich, Germany), and maintained in a humidified incubator with 5% CO_2_ at 37 °C. Cells were stimulated with the tested compounds (0.4, 2, 10, 25, and 50 μΜ) for 24 h after reaching 85–90% confluence. The cell number was determined in the presence of Trypan blue using an automated cell counting device (Thermo Fisher Scientific, Inc., Waltham, MA, USA).

### 4.3. Cell Viability Assessment

*Alamar Blue Assay.* The Alamar Blue colorimetric test was applied to determine the cell viability percentage of HaCaT and A375, after stimulation with increasing concentrations (0.4, 2, 10, 25, and 50 μΜ) of three triazole derivatives of betulinic (**1**), oleanolic (**2**), and ursolic (**3**) acids for 24 h. The cells (1 × 10^4^ cells/well) were seeded onto 96-well plates and incubated at 37 °C and 5% CO_2_, until reaching 85–90% confluence. The old medium was removed using an aspiration station and then replaced with a fresh medium containing the five tested concentrations of each phytocompound derivative. Some wells were also stimulated with the highest concentration of each triterpenic acid (50 μM). The test concentrations were obtained from three stock solutions of 10 mM and the final concentration of DMSO did not exceed 0.5%. After 24 h, the cells were further counterstained with Alamar Blue 0.01% and incubated for another 3 h. To quantify the cell population, the absorbance of the wells was then measured using a microplate reader (xMark^TM^ Microplate, Bio-Rad Laboratories, Hercules, CA, USA) at two wavelengths (570 nm and 600 nm).

*Cellular morphology evaluation*. The effect of **1**, **2,** and **3** on cellular morphology was assessed immediately after cell treatment (0 h) and at 24 h post-treatment. The images were captured and analyzed using the cellSens Dimensions v.1.8. Software (Olympus, Tokyo, Japan).

### 4.4. Immunofluorescence Assay—Morphological Assessment of Apoptotic Cells

The evaluation of nuclear localization and nuclear changes indicative of apoptosis (i.e. nuclear shrinkage/fragmentation) was performed using the 4,6′-diamidino-2-phenylindole (DAPI) Staining. HaCaT and A375 cells were seeded in 6-well plates at an initial cell density of 1 × 10^6^ cells/well. After cells reached 85–90% confluence, the old medium was removed and replaced with a fresh medium containing the highest concentrations of OA, UA, and BA derivatives (**1**, **2,** and **3** 50 μM) and BA, OA, and UA, respectively. The cells were treated with the tested compounds and incubated for 24 h at 37 °C. After the incubation period, the staining protocol was performed through the following steps: the cells were washed 2–3 times with cold phosphate-buffered saline PBS (1X) (Thermo Fisher Scientific, Boston, MA, USA), fixed up with paraformaldehyde 4% in PBS and then permeabilized with Triton X/PBS 2% for 30 min at room temperature. After another washing step with cold PBS, the cells were blocked with 30% FCS in 0.01% Triton X for 1 h at room temperature. In the end, the cells were washed again 2–3 times with cold PBS, stained with a solution of 4,6′-diamidino-2-phenylindole (DAPI, 300 nM), and incubated at 4 °C in the dark overnight. The nuclear alterations were analyzed using the integrated DP74 digital camera of an Olympus IX73 inverted microscope (Olympus, Tokyo, Japan).

### 4.5. Quantitative Real-Time PCR

Total RNA was isolated using the TRIzol reagent (Thermo Fisher Scientific, Inc., Waltham, MA, USA) and the Quick-RNA™ purification kit (Zymo Research Europe, Freiburg im Breisgau, Germany). Total RNA was further transcribed was conducted with Maxima^®^ First Strand cDNA Synthesis Kit ( Thermo Fisher Scientific, Inc. Waltham, MA, USA). Quantitative real-time PCR analysis was conducted by the means of Quant Studio 5 real-time PCR system (Thermo Fisher Scientific, Inc. Waltham, MA, USA) in the presence of Power SYBR-Green PCR Master Mix (Thermo Fisher Scientific, Inc.). The primer pairs (Thermo Fisher Scientific, Inc. Waltham, MA, USA) used are listed in Table 2. Normalized, relative expression data were calculated using the comparative threshold cycle (2^−∆∆Ct^) method.

### 4.6. High-Resolution Respirometry

The mitochondrial respiratory function was assessed using high respirometry studies (Oxygraph-2k Oroboros Instruments GmbH, Innsbruck, Austria) at 37 °C. To assess the mitochondrial respiratory rates of permeabilized HaCaT and A375 cells, a modified substrate uncoupler–inhibitor titration (SUIT) protocol was followed to obtain both separate and convergent Complex I and Complex II (CI + CII) electron input, as previously described by Petruș et al. [55]. Prior to mitochondrial function evaluation, the cells were cultured in T75 culture flasks, treated with the tested compounds (**1, 2,** and **3**) for 24 h, washed with PBS, trypsinized, counted, and resuspended (1 × 10^6^/mL) in mitochondrial respiration medium (MIRO5: MgCl_2_ 3 mM, EGTA 0.5 mM, taurine 20 mM, KH_2_PO_4_ 10 mM, K-lactobionate 60 mM, D-sucrose 110 mM, HEPES 20 mM, and BSA 1 g/L, pH 7.1). The concentrations used were those at which the tested compounds produced a significant decrease of A375 melanoma cell viability without decreasing that of HaCaT healthy cells: 25 μM for compounds **1** and **2** and 50 μM for compound **3**.

The cellular membrane was permeabilized using a mild detergent: digitonin (35 μg/L × 10^6^ cells) to evaluate the extended functional oxidative phosphorylation (OXPHOS) and to allow the soluble molecule to pass between external media and cytosol. The suitable digitonin concentration for selective plasma membrane permeabilization was determined in a previous respirometry protocol by step-wise titrations, as described by Pesta and Gnaiger [56]. The first respiratory rate recorded was routine respiration, obtained after the cells were suspended in free media, for 15 min. After the routine measurement, the subsequently applied SUIT protocol consisted in the addition of the substrates, as follows: (i) digitonin (cells permeabilizer) and the CI substrates (glutamate (G), 10 mM and malate (M), 5 mM) to measure basal respiratory rate (State 2_CI_), (ii) ADP (5 mM) to measure the active respiration dependent on CI (OXPHOS_CI_), (iii) succinate-S (10 mM), a C II substrate, which induced maximal OXPHOS capacity of both CI and CII (OXPHOS_CI+CII_), (iv) oligomycin (1 μg/mL), an inhibitor of CV used to measure the basal respiration dependent on both CI and CII (State 4_CI+II_), (v) *p*-(trifluoromethoxy) phenylhydrazone carbonyl cyanide-FCCP (1 μM/step) successive titrations to determine the maximal respiratory capacity of the electron transport system (ETS_CI+II_), (vi) rotenone (0.5 μM), a CI inhibitor used to determine the maximal respiratory capacity of the electron transport system dependent solely on CI (ETS_CI_), and (vii) antimycin A (2.5 μM)—a complex CIII inhibitor, to completely inhibit the electron transport system and measure the residual oxygen consumption (ROX). All the obtained values were corrected after ROX.

All data were recorded using DatLab software (Oroboros Instruments GmbH, Innsbruck, Austria) and analyzed using the GraphPad Prism 5 software (GraphPad Software, Inc., San Diego, CA, USA). Statistically significant differences were compared using a two-way analysis of variance (ANOVA) with Bonferroni’s post hoc test. Values with *p* < 0.05 were considered to have statistically significant differences (* *p* < 0.05, ** *p* < 0.01, and *** *p* < 0.01).

### 4.7. Molecular Docking

The molecular docking protocol used in our current work was previously described [57,58]. The protein target structure of Bcl-2 (PDB ID: 2W3L) was obtained from the RCSB Protein Data Bank [59]. The protein file was optimized using the AutoDock Tools v1.5.6 software package (The Scripps Research Institute, La Jolla, CA, USA). Water molecules and the native co-crystallized ligand were removed, after which Gasteiger charges were added to the protein. Structures of compounds **1**–**3**, BA, UA, and OA were sketched using Biovia Draw (Dasault Systems Biovia, San Diego, CA, USA) and subsequently converted into 3D structures using the Open Babel module (UFF force field) from PyRx v0.8 (The Scripps Research Institute, La Jolla, CA, USA). Molecular docking was conducted with PyRx v0.8 (The Scripps Research Institute, La Jolla, CA, USA) using AutoDock Vina’s available scoring function [60]. The docking protocol was validated by docking the native co-crystallized ligand into its original binding pocket. The calculated docking pose was compared with the experimental available binding pose. Docking studies were performed only if the root mean square deviation (RMSD) values between the two above-mentioned poses did not exceed a 2 Å threshold. Coordinates for the grid box were, center x = 37.5786 Å, y = 27.8119 Å, z = −9.6370 Å; size x = 22.7992 Å, y = 20.6680 Å, z = 21.1332; exhaustiveness = 8. Docking scores were recorded as ∆G binding energy values (kcal/mol). Compound–protein interactions were analyzed and depicted using Accelrys Discovery Studio 4.1 (Dassault Systems Biovia, San Diego, CA, USA).

## 5. Conclusions

The current study reports the synthesis, characterization, and antiproliferative biological assessment of the benzotriazolyl esters of BA (1), OA (2), and UA (3) against human malignant melanoma. The three compounds were synthesized by condensing HOBt with the COOH group of the triterpenic acids, using DCC as a reaction promoter. Compound identity was validated through 1H- and 13C-NMR, FTIR, and LC-MS. The cytotoxic activities of the esters and parent compounds were subsequently tested on A375 human melanoma cells and HaCaT healthy keratinocytes. Compounds 1 and 2 showed increased cytotoxic activity against melanoma cells while compound 3 reduced cell viability to a lesser degree. Furthermore, all compounds showed no cytotoxicity against healthy cells (HaCaT). DNA staining of treated cells revealed that the esters induce morphological changes in melanoma cells consistent with apoptosis. These changes are backed up by rtPCR results that highlight a reduction in antiapoptotic Bcl-2 expression and at the same time upregulation of the pro-apoptotic Bax. High-resolution respirometry studies revealed that all compounds were also able to significantly inhibit mitochondrial function. Molecular docking showed that the three esters show an increased affinity against Bcl-2 as compared to their triterpenic acid precursors. Moreover, compounds bind in the same hydrophobic region of the protein where the Bak, Bad, or Bim peptides bind. Taken together, we conclude that HOBt esterification is a viable method that increases the apoptotic effect of triterpenic acids in melanoma and can be used as a useful tool in the design of future triterpene-based semisynthetic compounds with augmented anticancer potential.

## Figures and Tables

**Figure 1 ijms-23-09992-f001:**
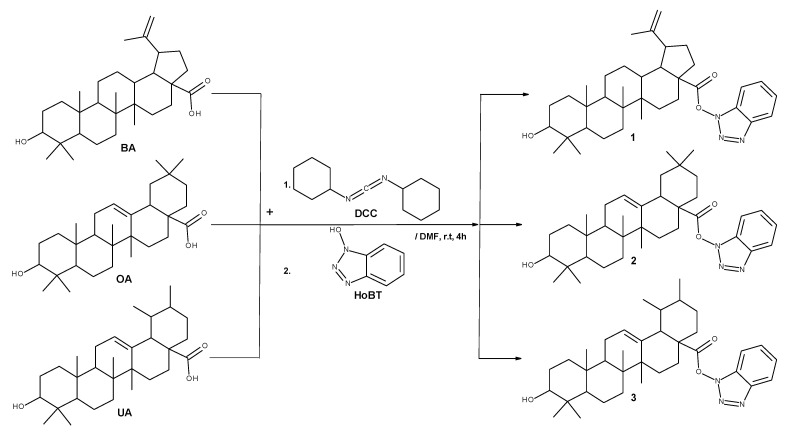
Synthesis procedure of BA, OA, and UA-HOBt esters (**1**–**3**).

**Figure 2 ijms-23-09992-f002:**
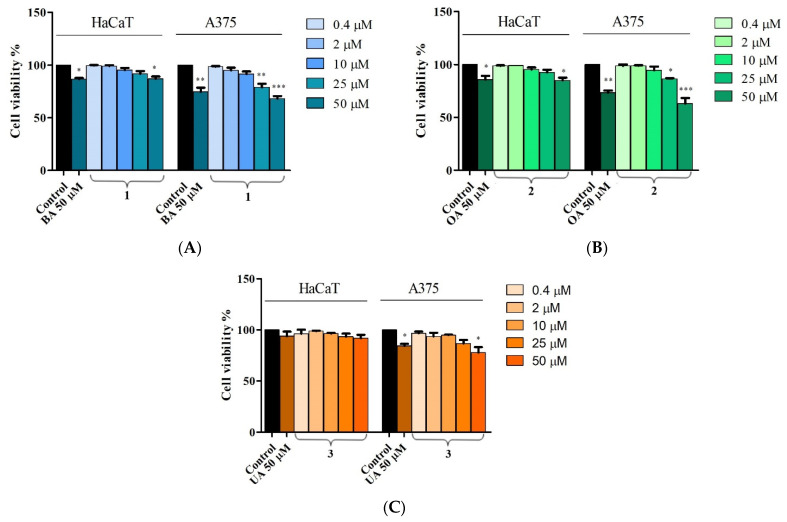
Cell viability of HaCaT and A375 cells after 24 h of treatment with 0.4, 2, 10, 25, and 50 μM of compounds **1** (**A**), **2** (**B**), and **3** (**C**), determined using the Alamar Blue assay. The results are expressed as cell viability percentage (%) normalized to control (100%). The data represent the mean values ± SD of three independent experiments performed in triplicate. The statistical differences vs. the control was determined using one-way ANOVA analysis followed by Tukey’s multiple comparisons post-test (* *p* < 0.05, ** *p* < 0.005 and *** *p* < 0.0001).

**Figure 3 ijms-23-09992-f003:**
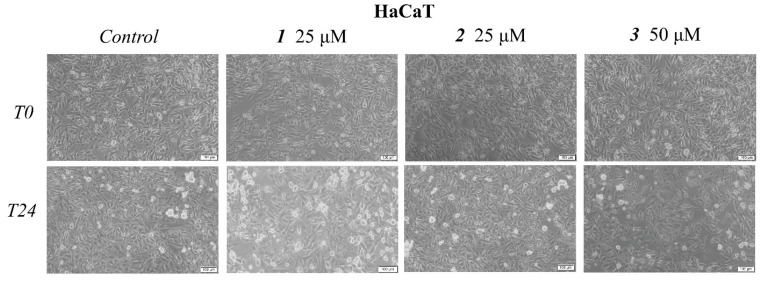
Representative images of the morphological aspects of HaCaT cells after treatment for 24 h with **1**, **2** (25 μM), and **3** (50 μM). The scale bar was 100 μm.

**Figure 4 ijms-23-09992-f004:**
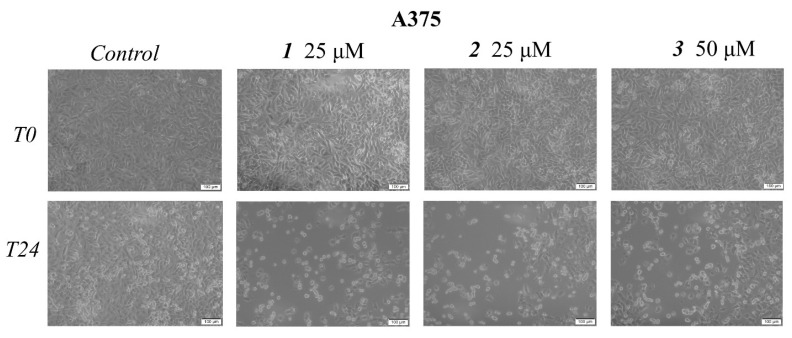
Representative images of the morphological aspects of A375 cells after treatment for 24 h with **1**, **2** (25 μM), and **3** (50 μM). The scale bar was 100 μm.

**Figure 5 ijms-23-09992-f005:**
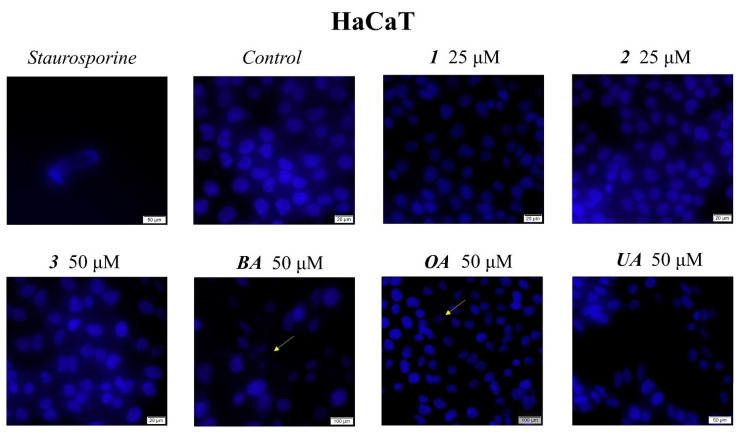
Nuclear staining using DAPI in HaCaT cells after treatment with **1**, **2**, and **3** (10, 25, and 50 μM) for 24 h. The pictures were captured 24 h post-treatment. The staurosporine solution (5 μM) was used as the positive control for apoptotic changes at the nuclear level. The yellow arrows represent signs of apoptosis, such as nuclear shrinkage, condensation, fragmentation, and cellular membrane disruption.

**Figure 6 ijms-23-09992-f006:**
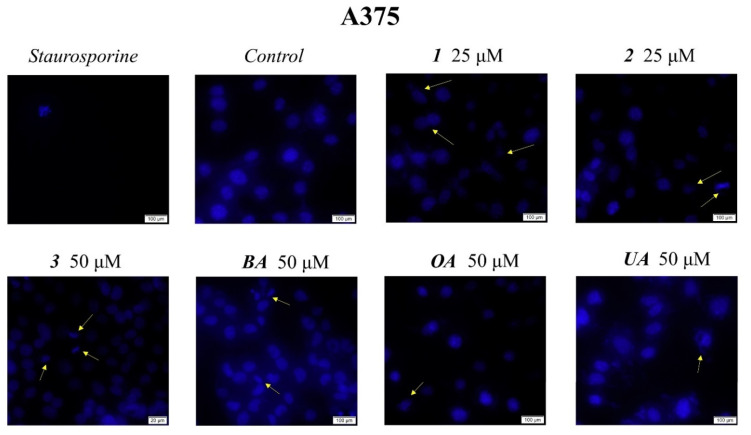
Nuclear staining using DAPI in A375 cells after treatment with **1**, **2**, and **3** (10, 25, and 50 μM) for 24 h. The pictures were captured 24 h post-treatment. The staurosporine solution (5 μM) was used as the positive control for apoptotic changes at the nuclear level. The yellow arrows represent signs of apoptosis, such as nuclear shrinkage, condensation, fragmentation, and cellular membrane disruption.

**Figure 7 ijms-23-09992-f007:**
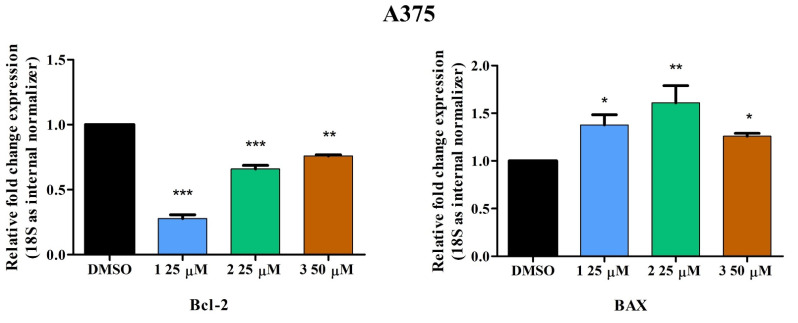
Relative fold change expression in mRNA of Bcl-2 and BAX in A375 cells after stimulation with **1, 2** (25 μM), and **3** (50 μM) for 24 h. The expressions were normalized to 18S and DMSO was used as the control. Data represent the mean values ± SD of three independent experiments. One-way ANOVA with Dunnett’s post-test was applied to determine the statistical differences in rapport with DMSO stimulated cells (* *p* < 0.05, ** *p* < 0.01, and *** *p* < 0.001).

**Figure 8 ijms-23-09992-f008:**
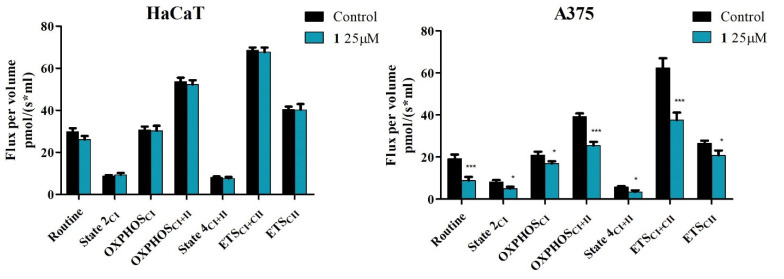
Respiration of permeabilized immortalized human keratinocytes (HaCaT) and human melanoma cells (A375) following a 24 h stimulation with 25 μM **1**. Data represent the mean ± SD of five individual experiments. Values with *p* < 0.05 were considered to have statistically significant differences (* *p* < 0.05, and *** *p* < 0.01). The respiratory parameters displayed represent the following: Routine—respiration of cells suspended in a substrate-free media, supported by endogenous ADP; State 2_CI_—mitochondrial respiration in basal conditions driven by CI, OXPHOS_CI_—active respiration dependent on CI substrates and exogenous ADP; OXPHOS_CI+II_—maximal active respiration driven by both CI and CII; State 4_CI+II_—basal respiration dependent on both CI and CII; ETS_CI+II_—maximal respiratory capacity of the electron transport system in the fully noncoupled state; ETS_CII_—electron transport system maximal capacity dependent only on CII.

**Figure 9 ijms-23-09992-f009:**
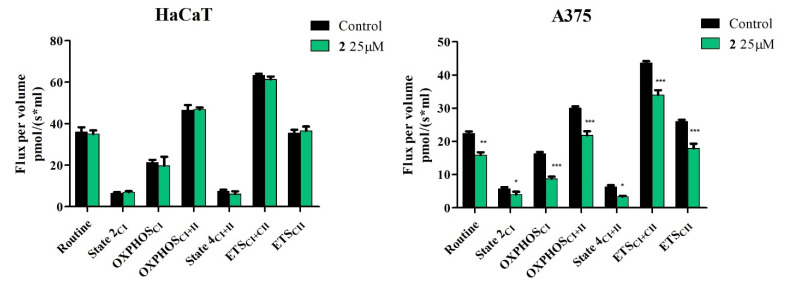
Respiration of permeabilized immortalized human keratinocytes (HaCaT) and human melanoma cells (A375) following 24 h stimulation with 25 μM **2**. Data represent the mean ± SD of five individual experiments. Values with *p* < 0.05 were considered to have statistically significant differences (* *p* < 0.05, ** *p* < 0.01, and *** *p* < 0.01). The respiratory parameters displayed represent the following: Routine—respiration of cells suspended in a substrate-free media, supported by endogenous ADP; State 2_CI_ –mitochondrial respiration in basal conditions driven by CI, OXPHOS_CI_—active respiration dependent on CI substrates and exogenous ADP; OXPHOS_CI+II_—maximal active respiration driven by both CI and CII; State 4_CI+II_—basal respiration dependent on both CI and CII; ETS_CI+II_—maximal respiratory capacity of the electron transport system in the fully noncoupled state; ETS_CII_—electron transport system maximal capacity dependent only on CII.

**Figure 10 ijms-23-09992-f010:**
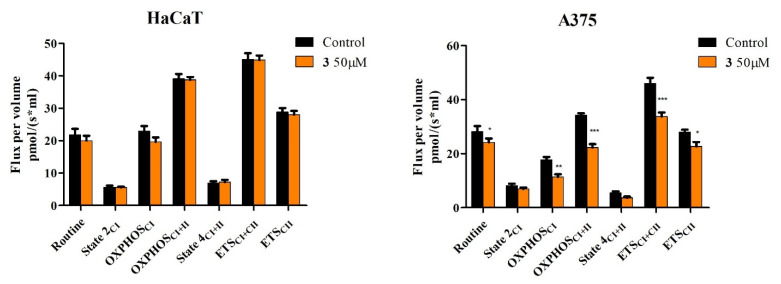
Respiration of permeabilized immortalized human keratinocytes (HaCaT) and human melanoma cells (A375) following 24 h of stimulation with 50 μM **3**. Data represent the mean ± SD of five individual experiments. Values with *p* < 0.05 were considered to have statistically significant differences (* *p* < 0.05, ** *p* < 0.01, and *** *p* < 0.01). The respiratory parameters displayed represent the following: Routine—respiration of cells suspended in a substrate-free media, supported by endogenous ADP; State 2_CI_—mitochondrial respiration in basal conditions driven by CI, OXPHOS_CI_—active respiration dependent on CI substrates and exogenous ADP; OXPHOS_CI+II_—maximal active respiration driven by both CI and CII; State 4_CI+II_—basal respiration dependent on both CI and CII; ETS_CI+II_—maximal respiratory capacity of the electron transport system in the fully noncoupled state; ETS_CII_—electron transport system maximal capacity dependent only on CII.

**Figure 11 ijms-23-09992-f011:**
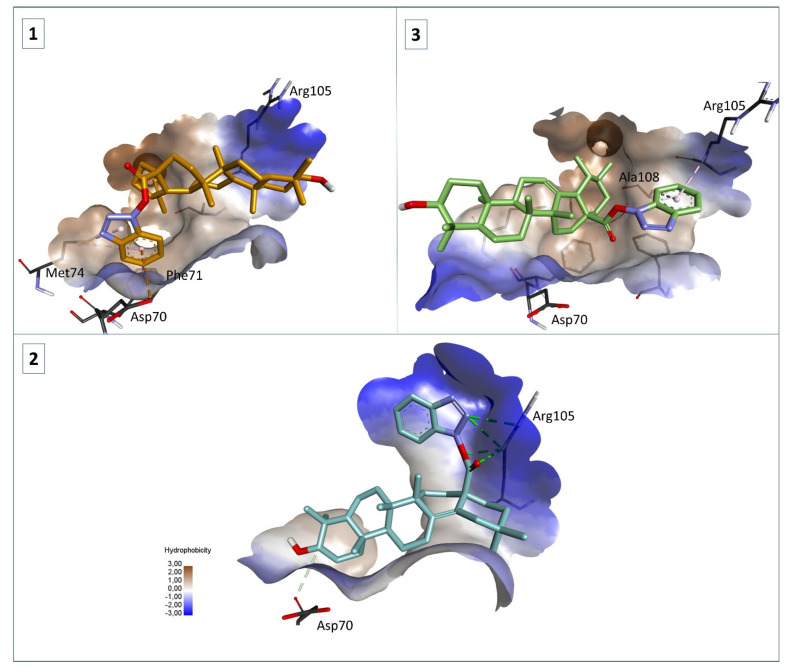
Structure of Bcl-2 (2W3L) in complex with compounds **1** (orange), **2** (blue), and **3** (green); HB interactions are depicted as green dotted lines, hydrophobic interactions as purple dotted lines, and electrostatic interactions as orange dotted lines; interacting amino acids are shown as dark gray sticks.

**Table 1 ijms-23-09992-t001:** Bcl-2 docking scores for compounds **1**–**3**, BA, OA, and UA (binding energy, ∆G kcal/mol).

Docked Compound	∆G (kcal/mol) for Bcl-2 (PDB ID: 2W3L)
**NL**	−10.3
**1**	−9.0
**2**	−9.4
**3**	−9.0
BA	−7.3
OA	−7.7
UA	−7.3

NL—native ligand of 2W3L.

**Table 2 ijms-23-09992-t002:** Primer pair used in rtPCR analysis.

Sequence Name	Forward	Reverse
**18 S**	5′GTAACCCGTTGAACCCCATT 3′	5′CCA-TCC-AAT-CGG-TAGTAG-CG 3’
**BAX**	5′GGCCGGGTTGTCGCCCTTTT 3′	5′CCGCTCCCGGAGGAAGTCCA 3’
**Bcl-2**	5′CGGGAGATGTCGCCCCTGGT 3′	5′-GCATGCTGGGGCCGTACAGT-3′

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
