# Peer review of "Novel Triterpenic Acid—Benzotriazole Esters Act as Pro-Apoptotic Antimelanoma Agents"

_ijms, 2022, doi:10.3390/ijms23179992_

Round 1
Reviewer 1 Report
The article which I received for review describes the synthesis of three simple derivatives of triterpene acids (OA, UA, and BA) and a large set of biological and computational studies for the obtained three new compounds. Even though the work on the chemical and synthetic side is quite trivial, together with the described numerous biological and computational studies, it constitutes a very interesting and coherent presentation. It provides a lot of new knowledge on the pro-apoptotic activity of new triterpene compounds. This broad approach to the conducted research and the skilful combination of the chemical part with the biological part is a very precious look for a problem.
This work, both in terms of its form and content, meets all formal requirements to be recommended for publication in IJMS. This is especially important in the context of a referral it to the Special Issue of IJMS, entitled "Pentacyclic Triterpene 2". This work perfectly corresponds to the subject of this Special Issue.
Although the work is very good in its general nature, I have a few minor detailed remarks. These are:
1. Use of the name and its abbreviation for the substance "hydroxybenzotriazole”. I believe that when Authors giving the full name it should be" 1-hydroxybenzotriazole". Moreover, the commonly accepted form of the abbreviation for this compound is" HOBt ", or eventually" HOBT ". Any others, in my opinion, are incorrect, and above all, the use of this abbreviation must be standardized throughout all work, because currently it is used in various forms.
2. In my opinion, the two chapters: Introduction and Discussion, are treated too broadly and are simply too extensive. They contain several elements of great general importance, but not specific just to this special subject. I believe that for better readability of the work, these two chapters could be significantly shortened.
3. The work requires very careful and thorough editing correction. In the text there are many so-called typing errors and many inappropriate division and transfer of word fragments between individual lines.
Following these minor remarks, this work can be approved for publication in IJMS without re-review.
Author Response
We are grateful and highly appreciative, of the reviewer’s work. Their observation allowed us to develop an improved form of the manuscript. Here is our point by point detailed answer to the reviewers’ remarks and observations. All changes made to the revised form of the manuscript are done using the Word embedded “Track changes” option.
- Use of the name and its abbreviation for the substance "hydroxybenzotriazole”. I believe that when Authors giving the full name it should be" 1-hydroxybenzotriazole". Moreover, the commonly accepted form of the abbreviation for this compound is" HOBt ", or eventually" HOBT ". Any others, in my opinion, are incorrect, and above all, the use of this abbreviation must be standardized throughout all work, because currently it is used in various forms.
Answer: We thank the esteemed reviewer for his observation. We made all changes to the text, as advised.
- In my opinion, the two chapters: Introduction and Discussion, are treated too broadly and are simply too extensive. They contain several elements of great general importance, but not specific just to this special subject. I believe that for better readability of the work, these two chapters could be significantly shortened.
Answer: We thank the esteemed reviewer for the useful suggestion. The aforementioned sections were shortened. However, we added a paragraph to the discussion section, to clarify some mechanism of action-related aspects, as was requested by another reviewer.
- The work requires very careful and thorough editing correction. In the text there are many so-called typing errors and many inappropriate division and transfer of word fragments between individual lines.
Answer: We thank the reviewer for the suggestion. We thoroughly read the text and fixed all typing errors. Unfortunately, the inappropriate division and transfer of word fragments between individual lines is done by the IJMS template which does automated text formatting so it can’t be changed.
Reviewer 2 Report
In this work, the Authors report the chemical synthesis and characterization of benzotriazole esters of betulinic, ursolic, and oleanolic acids against human malignant melanoma. They also try to elucidate their mechanism of action correlated with the antiproliferative effect. Moreover, they employed molecular docking in order to analyze the binding affinity and interaction mode of the studied compounds against Bcl-2. The studies are well-conducted and -organized. All Figures are legible and correctly described. The Materials and Methods section provides enough details making the studies highly reproducible. The presented data are sufficiently sound to support the conclusions, although the subject requires further investigation. Thus, it is worth adding some information on the directions of future research. While I believe that it is an interesting work and the obtained results look promising, I noted some minor errors that should be corrected prior to publication.
Minor notes:
- There are a lot of typographic minor errors or unclear elements. I would suggest that the Authors go through the text once again and correct them: citotoxicity → cytotoxicity (line 38), anti-cancer → anticancer (line 93), Therefor → Therefore, (line 109), synthetised → synthetized (line 144), vs. → vs. (line 146), figures → Figures (line 159), the → The (line 184), appliedto → applied to (line 193), Agilent6120 → Agilent 6120 (line 527), HoBt → HoBT (line 539), antiapoptotic → anti-apoptotic (line 725).
- Line 68: Maybe ‘the most investigated effects by far were their anticancer and chemopreventive effects’ will be better.
- Line 96: I propose ‘pancreatic and breast cancers’.
- Lines 118, 661: 1-3 should be in bold as before.
- Line 118: DCC – explain the abbreviation at the first use. The same for BA, OA, UA (line 27), HoBT (line 119), ROS (line 247).
- Line 131: 24h – no spaces here and elsewhere.
- Decide RT-PCR (line 32) or rtPCR (lines 38, 724), upregulation (lines 33, 440, 726) or up-regulation (line 187)?
- Line 155: There should be Figure 3 first mentioned.
- Lines 242 – 244: ‘The decrease of State2CI and State4CI+CII reveals that only compounds 1 and 2 can lower the residual respiration that compensates for the proton leak in the non-phosphorylating resting state, when ATP synthase is not active…’ – I think that this sentence seems to apply to what is shown in Figures 8-9.
- Lines 160 – 161, 266: ‘Discussion’ instead of ‘Discussions’.
- Table 2: Explain the meaning for Seq, Fv, Rev.
- Line 659: Include appropriate reference number.
- Line 667: It should be ‘μg/L’.
With kind regards,
Reviewer
Author Response
We are grateful and highly appreciative, of the reviewer’s work. Their observation allowed us to develop an improved form of the manuscript. Here is our point by point detailed answer to the reviewers’ remarks and observations. All changes made to the revised form of the manuscript are done using the Word embedded “Track changes” option.
We made all the required changes to the revised version of the manuscript. We would like to add that HoBT was changed to HOBt as requested by another reviewer’s suggestion, stating that it is the officially accepted abbreviation and the introduction and discussion sections were also shortened as requested by another reviewer. We are again very appreciative of the reviewer’s extensive work and help.
Reviewer 3 Report
The authors presented a fairly high-quality study. They have been researching the effects of triterpenoids for a long time and seem to have a lot of experience in this area. I have only a small comment. In their works, the authors should also note the direct membranotropic effects of the studied compounds. Indeed, being generally hydrophobic, triterpenoids (such as betulin and its numerous derivatives) are able to accumulate in membranes and change their properties. Recent works show that this applies to cell membranes, mitochondrial membranes, and artificial membrane systems. And this, in general, limits their bioavailability. In particular, the authors permeabilized the cells when assessing the respiration of their mitochondria. On the one hand, this is a requirement of the method, and on the other hand, it allows triterpenoids to freely enter the cell and affect mitochondria. Therefore, the design of this experiment does not quite allow the authors to assess the possible effect on mitochondria in cell culture. It is preferable to evaluate the membrane potential of mitochondria using a cytofluorimeter without additional permeabilization, which affects the interpretation of the experimental results. Therefore, it can be assumed that cytotoxic effects may also be due (if not primarily) to membranotropic action. This should be reflected in the work.
Author Response
We are grateful and highly appreciative, of the reviewer’s work. Their observation allowed us to develop an improved form of the manuscript. Here is our point by point detailed answer to the reviewers’ remarks and observations. All changes made to the revised form of the manuscript are done using the Word embedded “Track changes” option.
Answer: We thank the reviewer for this observation. The reviewer is right in stating that mitochondrial membrane potential assessment would indeed suggest that permeabilization of the cells does not influence the effect of the compounds on mitochondria. However our institute is presently closed until September, and a new experiment with cells would take much longer than the time required for revising the manuscript.
Nevertheless in our experiment, the cell membranes were selectively permeabilized using digitonin at an optimal concentration (35 μg/l x 106 cells, determined in a previous respirometric protocol by step-wise titrations) in order to evaluate the extended functional oxidative phosphorylation (OXPHOS). However, in our study, the tested compounds were not added at the beginning of the respirometric protocol, but more precisely, the cells were treated for 24h with the respective compounds, prior to the respirometric protocol. Therefore, we consider that the effect of the newly triterpenoid derivatives on mitochondrial respiration was not a result of increased concentrations that reached mitochondria freely due to membrane permeabilization, as seen in protocols that follow the acute administration of compounds. Moreover, additional proof that the compounds are able to influence mitochondrial function, even before membrane permeabilization with digitonin, arises from the fact that all compounds were able to decrease the basal routine respiration, a mitochondrial respiratory rate recorded before digitonin addition. Our group recently studied the effect of betulinic acid on both mitochondrial function and membrane potential and the results showed similar inhibition of mitochondrial respiration similar to compounds 1-3, while the membrane potential was decreased. This work was also referend to in the text.
With regard to the membranotropic action of the compounds being responsible for the exhibited cytotoxicity, we concluded that this is not the primary cause because this effect is closely correlated with the hydrophobicity of the compounds, which is very similar. That could have translated into a cytotoxic effect of the compounds on healthy cells as well (like in other studies which were cited in the text), effect which is not present. Also compounds 2 and 3 are highly similar in terms of structure and hydrophobicity but have different effects on cells, thus showing that these effects are not solely influenced by the structures' hydrophobic nature alone. These details along with appropriate literature references were added to the text. We hope that these arguments presented in the text are clearly detailed and sufficient to please the reviewer’s expectations.